# Association of red and processed meat consumption with cancer incidence and mortality: An umbrella review protocol

Ying Li[1], Shuping Yang[1], Chenyu Yu[1], Mei Wu[1], Sibin Huang[1], Yong Diao[1], Xunxun Wu[1], Huiyong Yang[1]*, Zhenyu Ma[2,3]*

**1** School of Medicine, Huaqiao University, Quanzhou, Fujian, China, **2** Cheeloo College of Medicine, Shandong University, Jinan, China, **3** Department of Radiation Oncology, Shandong Cancer Hospital and Institute, Shandong First Medical University and Shandong Academy of Medical Sciences, Jinan, Shandong, China

* shyhy@hqu.edu.cn (HY); zhenyuma2001@163.com (ZM)

## Abstract

### Background

Many meta-analyses have reported the associations between red and processed meat consumption and cancer outcomes, but few have assessed the credibility of the evidence. In addition, the results of dose-effect analyses of the association between red and processed meat consumption and cancer outcomes were inconsistently reported in different articles. Here we propose a protocol for an umbrella review (UR) that be designed to assess these associations and explore the potential dose-response relationships.

### Methods

We will independently search five electronic databases and two registers from inception to July 2024 for systematic reviews with meta-analysis concerning the associations of red and processed meat consumption with cancer incidence and mortality. We will conduct the statistical analysis between August 2024 and December 2024. Also, an up-to-date search for additional primary studies of cancer outcomes that were not included in previously published meta-analyses will be conducted. The main outcomes will include the incidence and mortality of any cancer related to red and processed meat exposure. A series of unique associations will be created based on the cancer outcome, exposure, and clinical or population setting. For each association, we will update the meta-analysis by combining studies included in prior meta-analyses and new studies that were not included in prior meta-analyses, and re-perform the meta-analysis using the random-effects models. According to the credibility of the evidence assessment, all associations with a $P$ value of $\leq 0.05$ will be categorized as convincing, highly suggestive, suggestive, or weak evidence. All analyses will be performed in R (version 4.2.3).

### Results

The results of this UR are planned to be submitted to a peer-reviewed journal.

**Data availability statement:** No datasets were generated or analysed during the current study. All relevant data from this study will be made available upon study completion.

**Funding:** Natural Science Foundation of Fujian Province (Grant No.2022J05062 to Huiyong Yang); Huaqiao University Young and Middle-aged Teachers Science and Technology Innovation Funding Program (Grant No.ZQN-PY319 to Huiyong Yang); Natural Science Foundation of Xiamen (Grant No.3502Z20227041 to Huiyong Yang); and Scientific Research Funds of Huaqiao University (Grant No.21BS126 to Huiyong Yang).

**Competing interests:** The authors have declared that no competing interests exist.

**Abbreviations:** UR, Umbrella review; PRISMA-P, Preferred Reporting Items for Systematic review and Meta-Analysis Protocols; PROSPERO, International Prospective Register of Systematic Reviews; RCTs, randomized controlled trials; CSSs, cross-sectional studies; INPLASY, International Platform of Registered Systematic Review and Meta-analysis Protocols; AMSTAR 2, A Measurement Tool to Assess Systematic Reviews 2.

## Conclusion

The main aim of protocol publication is to get feed back from the reviewers to develop a standard protocol before its publication and after publication, it should guide this protocol to take up similar research by any researcher(s) by following meticulously this standard protocol.

## Registration

PROSPERO CRD42023414550.

## 1. Introduction

Red meat, which refers to the muscles of all kinds of mammals, such as beef, pork and mutton, and processed meat, which is the meat that has been processed by curing, salting, fermentation, smoking, using chemical preservatives and additives, or other treatments, are the important component of total dietary intake for many populations, with the increasing consumption globally [1,2]. Multiple studies have reported an increased risk of cancer (e.g., colorectal cancer, pancreatic cancer, breast cancer, bladder cancer, gastric cancer, non-Hodgkin lymphoma) associated with red and processed meat consumption [3–8]. In 2015, a working group of International Agency for Research on Cancer, concluded that red meat consumption is probably carcinogenic to humans, while processed meat is carcinogenic to humans after reviewing more than 800 articles on the relations between cancer and the consumption of red meat and processed meat [9].

Many systematic reviews of the contribution of red meat and processed meat products to cancer incidence and mortality have been published, but few have assessed the credibility of the evidence. In addition, the outcomes for different types of cancer were assessed in different studies and the results of dose-effect analyses of the relationship between red meat and processed meat consumption and cancer outcomes were inconsistently reported in different studies [10–12]. An umbrella review (UR) can be applied to summarize and grade the evidence from these meta-analyses with similar topics. Observational studies are more suitable for long-term follow-up as well as providing evidence from larger sample sizes than randomized controlled trials (RCTs), which is a better match for our study topic as cancer is less likely to occur as a result of short-term exposure to red and processed meat [13,14].

Hence, we plan to conduct an UR of observational studies on the basis of the following protocol, in an attempt to explore: (1) what are the cancers whose incidence and mortality are significantly associated with the consumption of red and processed meat? (2) how is the consumption of red and processed meat related to the incidence and mortality of these cancers and how is the dose-response effect? (3) what is the epidemiological estimation of various cancers in populations exposed to red and processed meat? So as to provide more reliable basis for people's dietary decisions.

## 2. Methods

This protocol followed the PRISMA-P (Preferred Reporting Items for Systematic review and Meta-Analysis Protocols) 2015 checklist [15], and was registered on PROSPERO (International Prospective Register of Systematic Reviews, No. CRD42023414550). The UR will be reported according to the S1 File: PRISMA checklist [16].

## 2.1. Search strategy

We will independently search five electronic databases (i.e., PubMed, Embase, Web of Science, Scopus, and the Cochrane Library) from inception to July 2024 for systematic reviews with meta-analysis of observational studies of the association between red/processed meat consumption and various cancers outcome. In addition, the PROSPERO and INPLASY (International Platform of Registered Systematic Review and Meta-analysis Protocols) also will be searched to identify potential studies. After all selected meta-analyses have been identified, an up-to-date search of the five electronic databases mentioned above and ClinicalTrials.gov for primary studies (i.e., cohort studies, case-control studies) of cancer outcomes will be conducted as an additional source of data. The comprehensive search strategy integrating medical subject headings with free-text terms will be used without the language and nationality restrictions. Table 1 presented a list of detailed search strategy for PubMed.

## 2.2. Selection criteria

### 2.2.1. Inclusion criteria.
(1) Type of participants: Meta-analyses of healthy or sick populations of any age, sex and ethnicity exposed or not exposed to red or processed meat will be included; (2) Type of exposure: Meta-analyses, measuring the exposure factors that are specified as red meat (i.e., pork, beef, mutton, or other mammalian meat) and processed meat products (red or white meat processed by curing, salting, fermentation, smoking, using chemical preservatives and additives, or other treatments), will be included [1]; (3) Type of

**Table 1. List of search strategy in PubMed.**

| Number | Search terms | |
|---|---|---|
| | **For systematic reviews** | **For updated primary studies*** |
| #1 | (((((((((((((((((meat[MeSH Terms]) OR (meat[Title/Abstract])) OR (red meat[Title/Abstract])) OR (meats[Title/Abstract])) OR (beef[Title/Abstract])) OR (lamb[Title/Abstract])) OR (mutton[Title/Abstract])) OR (goat[Title/Abstract])) OR (pork[Title/Abstract])) OR (processed meat[Title/Abstract])) OR (meat product[Title/Abstract])) OR (ham[Title/Abstract])) OR (sausage[Title/Abstract])) OR (hamburger[Title/Abstract])) OR (salami[Title/Abstract])) OR (bacon[Title/Abstract])) OR (processed meat[Title/Abstract])) OR (pastrami[Title/Abstract]) | (((((((((((((((((meat[MeSH Terms]) OR (meat[Title/Abstract])) OR (red meat[Title/Abstract])) OR (meats[Title/Abstract])) OR (beef[Title/Abstract])) OR (lamb[Title/Abstract])) OR (mutton[Title/Abstract])) OR (goat[Title/Abstract])) OR (pork[Title/Abstract])) OR (processed meat[Title/Abstract])) OR (meat product[Title/Abstract])) OR (ham[Title/Abstract])) OR (sausage[Title/Abstract])) OR (hamburger[Title/Abstract])) OR (salami[Title/Abstract])) OR (bacon[Title/Abstract])) OR (processed meat[Title/Abstract])) OR (pastrami[Title/Abstract]) |
| #2 | (((((((((((((((((((((neoplasms[MeSH Terms]) OR (neoplasm[Title/Abstract])) OR (cancer[Title/Abstract])) OR (carcinoma[Title/Abstract])) OR (tumor[Title/Abstract])) OR (tumour[Title/Abstract])) OR (malignant neoplasm[Title/Abstract])) OR (Malignancy[Title/Abstract])) OR (Malignancies[Title/Abstract])) OR (Neoplasm, Malignant[Title/Abstract])) OR (breast cancer[Title/Abstract])) OR (colorectal cancer[Title/Abstract])) OR (endometrial cancer[Title/Abstract])) OR (esophageal cancer[Title/Abstract])) OR (extrahepatic cancer[Title/Abstract])) OR (gallbladder cancer[Title/Abstract])) OR (liver cancer[Title/Abstract])) OR (ovarian cancer[Title/Abstract])) OR (pancreatic cancer[Title/Abstract])) OR (prostate cancer[Title/Abstract])) OR (gastric cancer[Title/Abstract])) OR (uterine cancer[Title/Abstract])) OR (lymphoma[Title/Abstract]) | (((((((((((((((((((((neoplasms[MeSH Terms]) OR (neoplasm[Title/Abstract])) OR (cancer[Title/Abstract])) OR (carcinoma[Title/Abstract])) OR (tumor[Title/Abstract])) OR (tumour[Title/Abstract])) OR (malignant neoplasm[Title/Abstract])) OR (Malignancy[Title/Abstract])) OR (Malignancies[Title/Abstract])) OR (Neoplasm, Malignant[Title/Abstract])) OR (breast cancer[Title/Abstract])) OR (colorectal cancer[Title/Abstract])) OR (endometrial cancer[Title/Abstract])) OR (esophageal cancer[Title/Abstract])) OR (extrahepatic cancer[Title/Abstract])) OR (gallbladder cancer[Title/Abstract])) OR (liver cancer[Title/Abstract])) OR (ovarian cancer[Title/Abstract])) OR (pancreatic cancer[Title/Abstract])) OR (prostate cancer[Title/Abstract])) OR (gastric cancer[Title/Abstract])) OR (uterine cancer[Title/Abstract])) OR (lymphoma[Title/Abstract]) |
| #3 | ((((((Meta-Analysis[Publication Type]) OR (Systematic Review[Publication Type])) OR (Meta-Analysis[Title/Abstract])) OR (Systematic Review[Title/Abstract])) OR (pooled analysis[Title/Abstract])) OR (synthesis[Title/Abstract]) | ((((((((((cohort study[MeSH Terms]) OR (case control study[MeSH Terms])) OR (cohort study[Title/Abstract])) OR (case control study[Title/Abstract])) OR (prospective study[Title/Abstract])) OR (retrospective study[Title/Abstract])) OR (observational study[Title/Abstract])) OR (longitudinal study[Title/Abstract])) OR (cohort[Title/Abstract])) OR (case control[Title/Abstract])) OR (case-control[Title/Abstract]) |
| #4 | #1 AND #2 AND #3 | #1 AND #2 AND #3 |

*New primary studies that that were not included in the previously published meta-analys.

outcomes: Meta-analyses that quantitatively assessed any outcome measures concerning the incidence and mortality of cancers will be included; (4) Type of studies: Systematic reviews that only included cohort studies and case-control studies and excluded RCTs and cross-sectional studies (CSSs) are eligible. All eligible systematic reviews are supposed to have been published in peer-reviewed journals.

**2.2.2. Overlapping studies.** Overlapping studies are defined as meta-analyses with the same cancer outcome, exposure, and clinical or population setting [17]. Inclusion of results from reviews with overlapping associations may lead to double inclusion of primary studies and bias in findings and estimates [18,19]. Therefore, where overlapping studies exist, we will select the one with the largest sample size [20–22].

**2.2.3. Exclusion criteria.** The two researchers will independently sift the retrieved records based on title, abstract, and full text. Following the PRISMA 2020 statement, the flow diagram of study selection will be shown as Fig 1. We will exclude the following studies: (1) meta-analyses of RCTs and CSSs; (2) meta-analyses of animal studies; (3) single arm meta-analyses without comparison groups; (4) meta-analyses of individual participant data; (5) meta-analyses of cancer risk predictive studies using meat consumption as a risk factor; (6) studies with pooled-analysis but with no systematic analysis methods; (7) meta-analyses without available data for evidence synthesis; (8) meta-analyses of gene polymorphism.

## 2.3. Study selection and data extraction

The EndNote software will be used to perform the data administration. Two investigators will independently screen the retrieved records for eligibility on the basis of the titles, abstracts and full texts. For each eligible meta-analysis, these two investigators will independently abstract the following information: title, first author name, year of publication, red and processed meat consumption, characteristics of participants, follow-up (for cohort studies), cancer outcomes, effect estimates and corresponding 95% confidence intervals, number and method-ological quality assessment of included individual studies, sample size, funding sources. The basic characteristics of each meta-analysis will be summarized as in the example in Table 2, and the Fig 2 provided a non-exhaustive summary of the analysis results from these sample meta-analyses [4,6–8]. The reported non-linearity *P* values will also be extracted when the meta-analysis considers dose-response effects. For each primary study (i.e., each individual study included in eligible meta-analyses and each updated primary study), they will also inde-pendently record the following data: unique identifier of each article (DOI), first author, pub-lication year, journal, study design, number of events and participants in each arm. Cohen's kappa coefficient will be calculated to assess consistency [23], and all inconsistencies will be resolved through discussion with a third investigator.

## 2.4. Data synthesis and analysis

Based on the cancer outcome, exposure, and clinical or population setting, a series of unique associations will be created. For each association, we will update the meta-analysis by combin-ing studies included in prior meta-analyses and new studies that were not included in prior meta-analyses, and re-perform the meta-analysis using the random-effects models for further standardized comparisons [17,24,25]. $I^2$ statistic will be used to evaluate the heterogeneity, and $I^2$ of more than 50% demonstrates the presence of significant heterogeneity [26]. The 95% prediction intervals will be calculated to estimate the potential range of fall in effect size of further studies [27]. Egger's regression asymmetry test and the comparison of random-effects summary effect size and that of the largest study will be applied to assess the small-study effect bias [17,28]. The chi-square test was used to compare the observed number of statistically

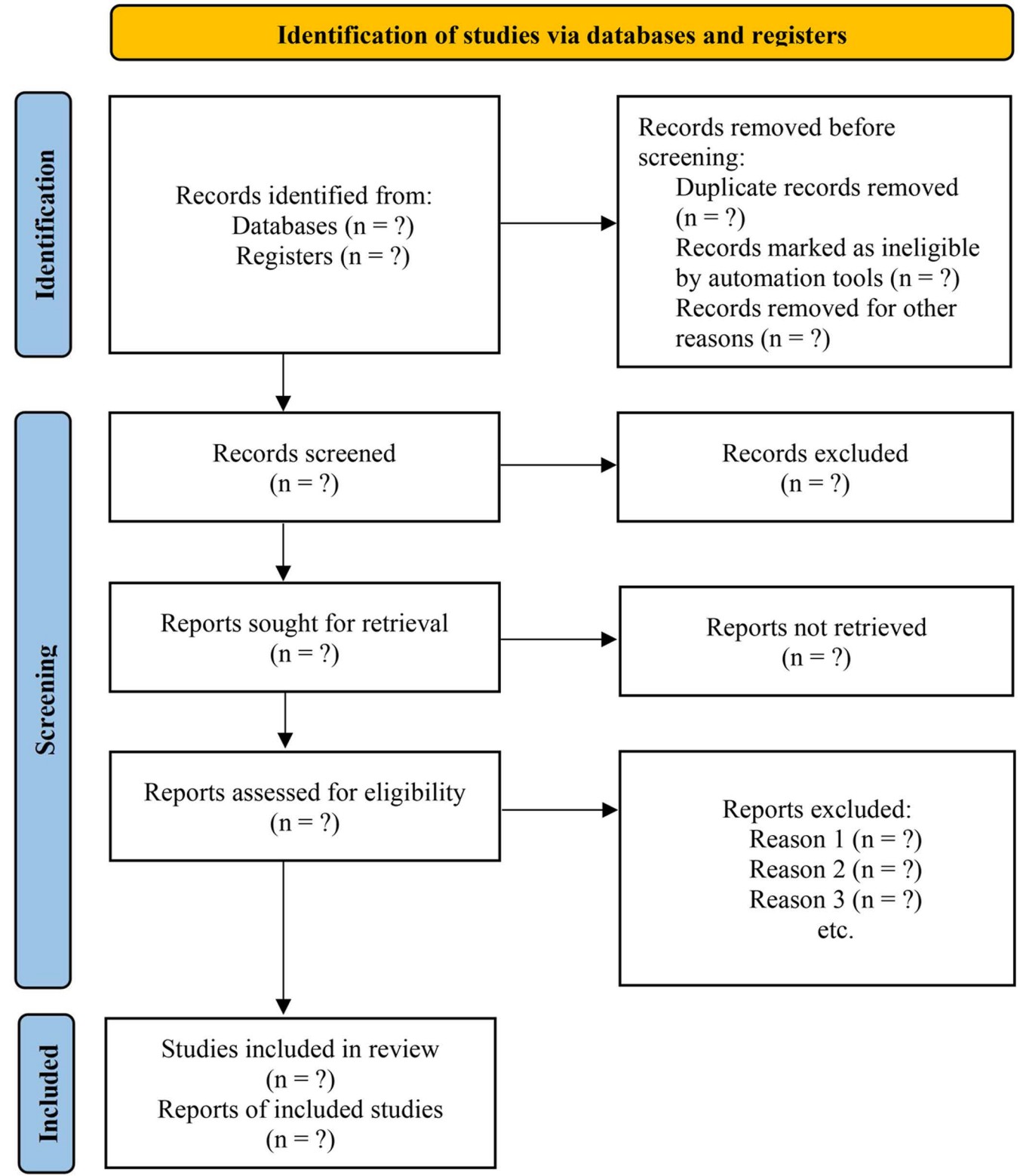

**Fig 1. Flowchart of the study selection process.**

**Table 2. Example of the list of some meta-analyses on the association between red meat and processed meat consumption and cancer risk.**

| Refer-ence | Publi-cation year | Databases searched in SR | Search dates | Follow-up (for cohort studies) | Sample size | Exposure details | Type of cancer | Outcome measures | Number of included studies | Method-ological quality assess-ment tool | Sub-group analysis | Sensi-tivity analysis | Dose-response analysis | Publica-tion bias assessment | Fund-ing source |
|---|---|---|---|---|---|---|---|---|---|---|---|---|---|---|---|
| Yang et al.'s SR [7] | 2015 | Medline and Embase | Inception to January 2015 | NA | 998,674 | Red meat; Processed meat; Bacon; Beef | Non-Hodgkin Lymphoma | Cancer risk | 20 (4 cohort studies, 16 case-control studies) | Newcastle-Ottawa Scale | Yes | Yes | Yes | Contour enhanced funnel plot and the Egger's test | None |
| Larsson et al.'s SR [6] | 2012 | PubMed and Embase | Incep-tion to Novembe r 2011 | 5–20 yrs | 2,307,787 | Red meat; Processed meat; Beef; Pork; Lamb; Ham; Sausage | Pancreatic cancer | Cancer risk | 13 prospective studies | NA | No | Yes | Yes | Egger's test | Aca-demic |
| Crippa et al.'s SR [4] | 2018 | PubMed | Inception to July 2016 | 10–27 yrs | 1,076,298 | Red meat; Processed meat; Beef; Pork; Lamb; Bacon; Hamburger; Hot dog; Ham; Sausage | Bladder cancer | Cancer risk | 13 (5 cohort studies, 8 case-control studies) | NA | Yes | Yes | Yes | Egger's test | Aca-demic |
| Zhao et al.'s SR [8] | 2017 | PubMed and Embase | Inception to October 2016 | 2–20 yrs | 2,154,151 | Red meat; Processed meat; Ham; Sausage; Bacon; Beef; Liver; Meat sauce | Gastric cancer | Cancer risk | 42 (9 cohort studies, 33 case-control studies) | Newcastle-Ottawa Scale | Yes | Yes | Yes | Funnel plot, Begg's t-est and Egger's test | Aca-demic |

SR = Systematic review, NA = Not Applicable, yrs = years.

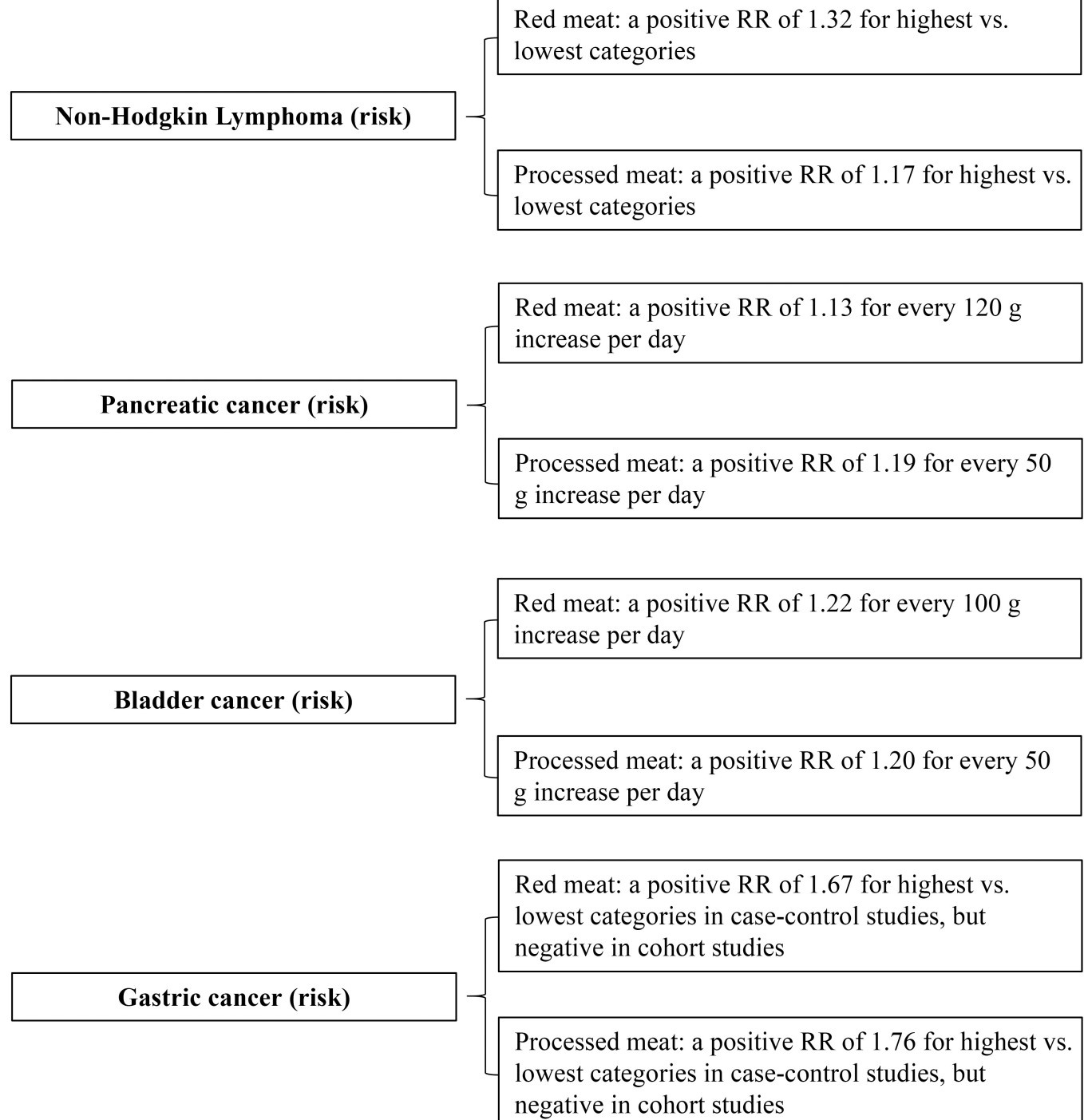

**Fig 2. A non-exhaustive summary of the analysis results from the sample meta-analyses.**

significant studies with the expected number of statistically significant studies to assess the excess significance bias [29,30]. For Egger's and excess significance tests, the level of significance will be set at $P = 0.10$; for other tests, the level will be $P = 0.05$. Recalculated dose-response analyses will also be performed if possible [31,32]. If quantitative synthesis is not

appropriate, descriptive analysis will be taken. The E-value methodology by VanderWeele and Ding will be used to assess how much of an unmeasured confounder would need to be associated with both the exposure (meat intake) and the outcome (risk and mortality of cancer) to adequately explain the currently estimated association [33]. The data concerning the incidence and mortality of various cancers related to red and processed meat consumption from cohort but not case-control studies will be summarized to provide an epidemiological estimation of the research findings. R software (version 4.2.2) will be used to perform all data analysis.

## 2.5. Methodological quality assessment

Two investigators will independently perform the methodological quality assessment of included meta-analyses using the AMSTAR (A Measurement Tool to Assess Systematic Reviews) 2, which contains seven critical domains and nine non-critical domains in total 16 items [34]. AMSTAR 2 can classify meta-analyses into high, moderate, low, and critically low confidence. The consistency of the two investigators' assessment results will also be assessed using kappa values, and inconsistencies will be resolved by discussing with a third investigator.

## 2.6. Credibility and quality of the evidence assessment

The credibility of the evidence of each association will be assessed on the basis of the following criteria: (1) associations with the number of participants of more than 1000, the $P$ value of $< 1 \times 10^{-6}$, the 95% PIs excluding the null, the largest study indicating a nominally significant effect size, the $I^2$ statistic of $< 50\%$, no evidence of small-study effects and excess significance biases will be classified as convincing evidence (class I); (2) associations with the number of participants of more than 1000, the $P$ value of $< 1 \times 10^{-6}$, and the $I^2$ statistic of $< 50\%$ will be classified as highly suggestive evidence (class II); (3) associations with the number of participants of more than 1000, the $P$ value of $< 1 \times 10^{-3}$ will be classified as suggestive evidence (class III); (4) all other associations only with the $P$ value of $\leq 0.05$ will be classified as weak evidence (class IV); (5) associations with the $P$ value of $> 0.05$ is non-significant evidence (NS) [35–37].

The quality of the evidence of each association will be assessed using the GRADE (Grading of Recommendations, Assessment, Development, and Evaluation) criteria [38]. The evidence from observational studies was initially considered to be of low quality per GRADE approach. Within the GRADE criteria, there are five dimensions (risk of bias, inconsistency, indirectness, inaccuracy, and publication bias) that can lead to a downgrading of the quality of the evidence, and three dimensions (confounders that may minimize the effect, dose-response, large magnitude of effect) that can lead to an upgrading of the quality of the evidence [38].

## 2.7. Sensitivity analysis

We will conduct sensitivity analyses on the exclusion of primary studies with sample sizes of more than 1000 to investigate the robustness of the results of meta-analyses that include smaller studies [39,40]. In addition, we will perform meta-analyses using fixed effects models for associations that include fewer than five studies [41].

## 2.8. Epidemiological estimate

Using this UR, we will be able to systematically quantify the prevalence and mortality of various cancers in populations exposed to red meat and processed meat after updating the latest primary studies (i.e., new published cohort studies and case-control studies that have not been included in pervious meta-analyses), and to further assess the risk difference for cancer

outcomes in populations with different consumption of red meat and processed meat by comparing the epidemiological estimate from our UR with incidence and mortality of various cancers reported in Global Cancer Statistics, 2020 [42]. Data from cohort studies rather than case-control studies will be summarized in order to make epidemiological estimates of study findings.

## 3. Discussion

Here, we propose an UR to assess the association of red and processed meat consumption with cancer outcomes and explore potential dose-response relationships. Although the results of many meta-analyses have shown that the consumption of red meat and processed meat is positively associated with an increased risk of many cancers, part of the studies differ in the definition of red meat and processed meat, study methodology, inclusion and exclusion criteria, and do not quantify the credibility of the evidence, which may have influenced the final evaluation and decision [4,6–8,43]. Therefore, this is an appropriate time to summarize the current evidence and quantify the credibility of the evidence based on uniform criteria.

To our knowledge, two URs concerning the association of red and processed meat consumption with cancer outcomes have been published [44,45]. However, the differences between our UR and these two URs are apparent from the comparison of the methodology shown in Table 3. By combining new primary studies not included in previously published meta-analyses, our UR will provide an up-to-date literature search and a more comprehensive evidence synthesis. Importantly, reevaluation of meta-analyses using the same effects model will facilitate more standardized comparisons and assessments of the summary effect sizes. The implementation of sensitivity analyses will allow further testing of the robustness of the evidence. In addition, we will conduct the epidemiological estimation of a series of cancer outcomes based on the data from included cohort studies.

According to this protocol, the cohort study is chosen to investigate the prevalence of various cancers in populations exposed to red and processed meat. However, the case-control study evaluating the effects of red and processed meat on cancer outcomes is based on the existing cases of cancer, and should not be considered for prevalence estimations. RCTs will not be considered because it is extremely difficult and costly to assess the long-term health effects of meat consumption through the RCTs.

The development of this study protocol will reduce statistical heterogeneity and allow investigators to further rigorously design and conduct this study. On the other hand,

**Table 3. Comparison of the methodology between the current UR and the other 2 URs.**

| Study | Database | Retrieval time | Eligible articles | Methodological quality assessment | Recalculation of summary effect | Sensitivity analysis | Epidemiological estimation of cancer outcomes |
|---|---|---|---|---|---|---|---|
| Huang et al.'s UR[44] | Medline, Embase, the Cochrane Database of Systematic Reviews, Web of Science | From the inception through January 2021 | Systematic reviews with meta-analysis | AMSTAR 1 | No | No | No |
| Grosso et al.'s UR[45] | Medline, Embase | From the inception through January 2017 | Systematic reviews with meta-analysis | NA | No | No | No |
| The current UR | PubMed, Embase, Web of Science, Scopus, the Cochrane Library, PROSPERO, INPLASY, ClinicalTrials.gov | From the inception through July 2024 | Systematic reviews with meta-analysis, Primary studies* (i.e., cohort studies, case-control studies) | AMSTAR 2 | Yes | Yes | Yes |

*New primary studies that that were not included in the previously published meta-analyses, NA = Not Applicable.

evaluating the associations of red and processed meat consumption with a broad range of cancer outcomes and exploring potential dose-response relationships will enable clinicians and health policy makers to make better recommendations on red meat and processed meat consumption.

## Supporting information

**S1 File. PRISMA Checklist** [16]**.**
(DOCX)

## Author contributions

**Conceptualization:** Ying Li, Huiyong Yang, Zhenyu Ma.

**Funding acquisition:** Huiyong Yang.

**Methodology:** Ying Li, Shuping Yang, Zhenyu Ma.

**Project administration:** Huiyong Yang.

**Writing – original draft:** Ying Li, Shuping Yang.

**Writing – review & editing:** Chenyu Yu, Mei Wu, Sibin Huang, Yong Diao, Xunxun Wu, Huiyong Yang, Zhenyu Ma.

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
