## [Decision Letter · Decision Letter 0]

6 Aug 2024

PONE-D-24-16182Association of red and processed meat consumption with cancer incidence and mortality: an umbrella review protocolPLOS ONE

Dear Dr. Yang,

Thank you for submitting your manuscript to PLOS ONE. After careful consideration, we feel that it has merit but does not fully meet PLOS ONE’s publication criteria as it currently stands. Therefore, we invite you to submit a revised version of the manuscript that addresses the points raised during the review process.

We look forward to receiving your revised manuscript.

Kind regards,

Ramy Mohamed Ghazy

Academic Editor

PLOS ONE

Journal Requirements:

"Natural Science Foundation of Fujian Province (Grant No.2022J05062 to Huiyong Yang); Huaqiao University Young and Middle-aged Teachers Science and Technology Innovation Funding Program (Grant No.ZQN-PY319 to Huiyong Yang); Natural Science Foundation of Xiamen (Grant No.3502Z20227041 to Huiyong Yang); and Scientific Research Funds of Huaqiao University (Grant No.21BS126 to Huiyong Yang)."

**Additional Editor Comments:**

Dear authors,

Based on the reviewer comments, I recommend minor revision. Kindly, accept my apologize for unintentional delay in the review process. Usually, finding qualified reviewers is not usually an easy task.

Reviewers' comments:

Reviewer's Responses to Questions

**Comments to the Author**

1. Does the manuscript provide a valid rationale for the proposed study, with clearly identified and justified research questions?

Reviewer #1: Yes

2. Is the protocol technically sound and planned in a manner that will lead to a meaningful outcome and allow testing the stated hypotheses?

Reviewer #1: Yes

3. Is the methodology feasible and described in sufficient detail to allow the work to be replicable?

Reviewer #1: Yes

4. Have the authors described where all data underlying the findings will be made available when the study is complete?

Reviewer #1: No

5. Is the manuscript presented in an intelligible fashion and written in standard English?

Reviewer #1: Yes

6. Review Comments to the Author

You may also provide optional suggestions and comments to authors that they might find helpful in planning their study.

Reviewer #1: Dear Authors

Thank you for providing me an opportunity to review your manuscript. The follwing are my comments/suggestions on the same:

Umbrella reviews are systematic collections and assessments of multiple SRMAs done on a specific research topic. The two most common applications of umbrella reviews deal with treatment effects of interventions and epidemiological associations of exposures. Umbrella reviews have to follow several steps like definition of research question. Development of search algorithm, definition of inclusion and exclusion criteria, systematic literature review, data extraction, statistical analysis of eligible meta-analysis, grading strength of evidence, sensitivity analysis, upgrade or down grade of ranking, and discussion of potential bias and related issues. In the figure one few of these logical sequences were missed.

Abstract

Methods: Protocols should always be a guiding document and it is like a standard operating procedure for following.

18: Why are authors mentioned to following 5 search engines and two registers. These should not be limited. Whatever is available, one can use.

19-20 The time period mentioned here is from inception to July 2024. When the inception occurs and what the total time (months/years) needs to be mentioned in the protocol.

27-29: According to the ………………….highly suggestive, suggestive (repetitive word may be deleted) or weak evidence.

Conclusions:

31-34: Conclusions need to be changed. The main aim of protocol publication is to get feed back/suggestions from the reviewers to develop a standard protocol before its publication and after publication, it should guide this protocol to take up similar research by any researcher(s) by following meticulously this standard protocol.

Main Manuscript

58-60: In addition ……………… in different studies. Please provide some references

96-99: This sentence is not clear, needs to be rewritten.

108: Sentence is not clear, needs to be rewritten.

114: Meta analysis ………………. evidence synthesis. It needs to be mentioned what is the cut point taken as the sufficiency of the data?

125-126: Please give reference to this sentence Fig 2.

Discussion: Maybe be fine pruned further.

Fig 2.

What is the basis of RR defined as highest or lowest for various cancer incidents versus consumption of meat and processed meat?

Methods: We will independently, …………………..from inception to 2024

7. PLOS authors have the option to publish the peer review history of their article (what does this mean? ). If published, this will include your full peer review and any attached files.

**Do you want your identity to be public for this peer review?** For information about this choice, including consent withdrawal, please see our Privacy Policy .

Reviewer #1: No

---

## [Author Response · Author response to Decision Letter 1]

16 Aug 2024

Umbrella reviews are systematic collections and assessments of multiple SRMAs done on a specific research topic. The two most common applications of umbrella reviews deal with treatment effects of interventions and epidemiological associations of exposures. Umbrella reviews have to follow several steps like definition of research question. Development of search algorithm, definition of inclusion and exclusion criteria, systematic literature review, data extraction, statistical analysis of eligible meta-analysis, grading strength of evidence, sensitivity analysis, upgrade or down grade of ranking, and discussion of potential bias and related issues. In the figure one few of these logical sequences were missed.

Response: Thank you very much for your comments. Figure 1 in the paper is a flow chart showing the literature search and selection, which is developed according to the official PRISMA website.

18: Why are authors mentioned to following 5 search engines and two registers. These should not be limited. Whatever is available, one can use.

Response:Thank you very much for your comments. The protocol of meta or umbrella analysis is an operational standard to improve the transparency and objectivity of meta-analysis. These 5 search engines and two registers were selected to define the operational process for our team to conduct meta-analysis in the future. According to AMSTAR-2, this is sufficient to identify the available literature.

19-20 Methods: We will independently, …………………..from inception to 2024

The time period mentioned here is from inception to July 2024. When the inception occurs and what the total time (months/years) needs to be mentioned in the protocol.

Response:Thank you very much for your comments. We have revised the article in detail according to your comments.

The time range here refers to the expected time range of literature retrieval. We expect the literature retrieval time to be from the establishment of the database to July 2024, while we add the statistical analysis time in the abstract: August 2024 to December 2024(page1, line 22-23). .

27-29: According to the ………………….highly suggestive, suggestive (repetitive word may be deleted) or weak evidence.

Response:Thank you very much for your comments. Highly suggestive and suggestive are standard terms based on evidence confidence evaluation criteria. Highly suggestive and suggestive represent two different levels.

Conclusions:

31-34: Conclusions need to be changed. The main aim of protocol publication is to get feed back/suggestions from the reviewers to develop a standard protocol before its publication and after publication, it should guide this protocol to take up similar research by any researcher(s) by following meticulously this standard protocol.

Response:Thank you very much for your comments. We have revised the article in detail according to your comments.

The conclusion of the article has been revised.

58-60: In addition ……………… in different studies. Please provide some references

Response:Thank you very much for your comments. We have revised the article in detail according to your comments.

We have added references 10-12 to the corresponding part of the article.

96-99: This sentence is not clear, needs to be rewritten.

Thank you very much for your comments. We have revised the article in detail according to your comments.

Response:We have changed the content of the article to: Systematic reviews that only included cohort studies and case-control studies and excluded RCTs and cross-sectional studies (CSSs) are eligible.

114: Meta analysis ………………. evidence synthesis. It needs to be mentioned what is the cut point taken as the sufficiency of the data?

Response:Thank you very much for your comments. We have revised the article in detail according to your comments.

We have changed the content of the article to: meta-analyses without available data for evidence synthesis.

125-126 Please give reference to this sentence Fig 2.

Fig 2.

What is the basis of RR defined as highest or lowest for various cancer incidents versus consumption of meat and processed meat?

Response:Thank you very much for your comments. We have revised the article in detail according to your comments.

We have given reference to this sentence Fig 2.

RR defined as highest or lowest for various cancer incidents is self-defined from the article in reference 8.

Discussion: Maybe be fine pruned further.

Response:Thank you very much for your comments. We have revised the article in detail according to your comments.

We have streamlined the discussion.

Journal Requirements:

2.We note that the grant information you provided in the ‘Funding Information’ and ‘Financial Disclosure’ sections do not match.

Please state what role the funders took in the study.

Response:Thank you for your comments. In the review comments, we add funders' help to the article, mainly including financial support for the use of search tools and the efficient operation of the team(page 18, line 244-256).

Comments to the Author

4.Have the authors described where all data underlying the findings will be made available when the study is complete?

Response:The data used in our article comes from public databases, i.e., PubMed, Embase, Web of Science, Scopus, and the Cochrane Library.

---

## [Decision Letter · Decision Letter 1]

26 Nov 2024

Association of red and processed meat consumption with cancer incidence and mortality: an umbrella review protocol

PONE-D-24-16182R1

Dear Dr. Huiyong Yang

We’re pleased to inform you that your manuscript has been judged scientifically suitable for publication and will be formally accepted for publication once it meets all outstanding technical requirements.

Kind regards,

Mehran Rahimlou, PhD

Academic Editor

PLOS ONE

Additional Editor Comments (optional):

Reviewers' comments:

Reviewer's Responses to Questions

**Comments to the Author**

1. Does the manuscript provide a valid rationale for the proposed study, with clearly identified and justified research questions?

Reviewer #1: Partly

Reviewer #2: Yes

2. Is the protocol technically sound and planned in a manner that will lead to a meaningful outcome and allow testing the stated hypotheses?

Reviewer #1: Yes

Reviewer #2: Yes

3. Is the methodology feasible and described in sufficient detail to allow the work to be replicable?

Reviewer #1: Yes

Reviewer #2: Yes

4. Have the authors described where all data underlying the findings will be made available when the study is complete?

Reviewer #1: No

Reviewer #2: Yes

5. Is the manuscript presented in an intelligible fashion and written in standard English?

Reviewer #1: Yes

Reviewer #2: Yes

6. Review Comments to the Author

You may also provide optional suggestions and comments to authors that they might find helpful in planning their study.

Reviewer #1: To

The Authors

I am thankful the Authors for their satisfactory revision. However, the Discussion part may need further improvement and need to bring more clarity in the discussion.

Reviewer #2: I truly appreciate the time, effort, and expertise you’ve put into this manuscript. Your manuscript addresses a significant area of study, and your contribution is valuable to advancing knowledge in the related fields

7. PLOS authors have the option to publish the peer review history of their article (what does this mean? ). If published, this will include your full peer review and any attached files.

**Do you want your identity to be public for this peer review?** For information about this choice, including consent withdrawal, please see our Privacy Policy .

Reviewer #1: **Yes: ** Laxmaiah Avula

Reviewer #2: No

---

## [Editor Report · Acceptance letter]

PONE-D-24-16182R1

PLOS ONE

Dear Dr. Yang,

I'm pleased to inform you that your manuscript has been deemed suitable for publication in PLOS ONE. Congratulations! Your manuscript is now being handed over to our production team.

Kind regards,

on behalf of

Dr. Mehran Rahimlou

Academic Editor

PLOS ONE